# Machine Learning Approximations to Predict Epigenetic Age Acceleration in Stroke Patients

**DOI:** 10.3390/ijms24032759

**Published:** 2023-02-01

**Authors:** Isabel Fernández-Pérez, Joan Jiménez-Balado, Uxue Lazcano, Eva Giralt-Steinhauer, Lucía Rey Álvarez, Elisa Cuadrado-Godia, Ana Rodríguez-Campello, Adrià Macias-Gómez, Antoni Suárez-Pérez, Anna Revert-Barberá, Isabel Estragués-Gázquez, Carolina Soriano-Tarraga, Jaume Roquer, Angel Ois, Jordi Jiménez-Conde

**Affiliations:** 1Neurovascular Research Group, Department of Neurology, IMIM-Hospital del Mar (Institut Hospital del Mar d’Investigacions Mèdiques), 08003 Barcelona, Spain; 2Unidad de Investigación AP-OSIs Guipúzcoa, 20014 Donostia, Spain; 3Medicine Department, DCEXS-Universitat Pompeu Fabra (UPF), 08002 Barcelona, Spain; 4Department of Psychiatry, NeuroGenomics and Informatics, Washington University School of Medicine, St. Louis, MO 63110, USA

**Keywords:** aging, epigenetic clock, machine learning, vascular risk factors, stroke

## Abstract

Age acceleration (Age-A) is a useful tool that is able to predict a broad range of health outcomes. It is necessary to determine DNA methylation levels to estimate it, and it is known that Age-A is influenced by environmental, lifestyle, and vascular risk factors (VRF). The aim of this study is to estimate the contribution of these easily measurable factors to Age-A in patients with cerebrovascular disease (CVD), using different machine learning (ML) approximations, and try to find a more accessible model able to predict Age-A. We studied a CVD cohort of 952 patients with information about VRF, lifestyle habits, and target organ damage. We estimated Age-A using Hannum’s epigenetic clock, and trained six different models to predict Age-A: a conventional linear regression model, four ML models (elastic net regression (EN), K-Nearest neighbors, random forest, and support vector machine models), and one deep learning approximation (multilayer perceptron (MLP) model). The best-performing models were EN and MLP; although, the predictive capability was modest (R^2^ 0.358 and 0.378, respectively). In conclusion, our results support the influence of these factors on Age-A; although, they were not enough to explain most of its variability.

## 1. Introduction

Age is the main risk factor for the most prevalent diseases in developed high-income countries [1]. The increase in life expectancy parallels also an increase in age-related diseases, such as cardiovascular disease, stroke, and dementia [2]. However, aging is a complex process that represents the decline of an organism at all levels, and there is a high inter-individual heterogeneity that is not explained by chronological age (C-Age) [3]. Since the early 1990s, there has been growing interest in finding new biomarkers that could represent the real or biological age (B-Age) of an individual [4].

One of the many biological mechanisms affected by aging is the change in DNA methylation (DNAm) throughout the lifespan. DNAm is an epigenetic mechanism that consists of the addition of a methyl group to the DNA without altering its sequence. This chemical modification most frequently occurs at cytosine-guanine dinucleotides (CpGs) and usually acts as a mechanism that regulates gene expression [5]. DNAm is a dynamic process that can be influenced by many internal and external factors, such as aging, lifestyle, or some diseases. Previous articles have described that aging affects DNAm, finding a specific DNAm signature for aging, such that some CpG sites become hypomethylated and others hypermethylated [6,7,8]. Promoted by the advances in microarray technologies [9,10,11], previous research used age-related changes in DNAm at specific CpGs to build B-age estimators, known as “epigenetic clocks” [12,13,14,15]. Therefore, the mismatch between C-Age and B-Age can be considered as an indicator of age acceleration (Age-A), which proved to predict a broad range of age-related health outcomes and mortality risks [16,17]. Regarding cerebrovascular diseases, we previously reported that B-Age is a better predictor than C-Age of stroke outcome, mortality, stroke recurrence, and white matter hyperintensity burden [18,19,20,21]. In addition, Age-A has been associated with a higher risk of cardiovascular diseases [22], earlier menopause [23], and frailty [24], confirming its usefulness as a marker of aging and biological health status.

Even if B-Age and Age-A are associated with relevant outcomes, their use in daily clinical practice is nowadays limited by the fact that DNAm data are not routinely collected. However, regardless of molecular and genetic factors that may contribute to the Age-A, the burden of environmental, lifestyle, and vascular risk factors (VRF) is known to affect Age-A [12,25]. For instance, a relationship has been described between Age-A and many of the following factors, namely, diet, tobacco use, alcohol consumption, air pollution, stress, insomnia, physical activity, or obesity [12,26].

Therefore, if Age-A is associated with these factors that can be easily measured, and if they can explain mostly Age-A, it might be possible to estimate Age-A based on them. To our best knowledge, no previous work has tried to estimate Age-A using highly available clinical data. Moreover, the rise in the number of epigenetic clocks has come together with the popularization of machine learning (ML) techniques. The first clocks used penalized linear regression models [13,14], but more recent B-Age models also use deep learning (DL) techniques [27,28], improving the C-Age predictions. Hence, it might be interesting to also check whether regular linear regression, ML, or DL models differ in the estimation of Age-A using clinical data.

The aim of this study is to estimate the contribution of easily measurable clinical factors to Age-A in patients with cerebrovascular disease (CVD), in order to investigate possible mechanisms involved in aging that could be modified, and also to find a more accessible model able to predict Age-A. Moreover, we will compare the performance in this Age-A estimation using different model-building approximations.

## 2. Results

First, we estimated Age-A (mismatch between B-Age and C-Age) using whole-blood DNAm data obtained from a cohort of individuals with acute CVD who had information on VRF, lifestyle habits, and target-organ damage (TOD). These variables were used as input in several ML prediction models of the Age-A, and we compared their performance in the validation stage. Predictor variables were previously centered to 0 (±1) and imputed if the missing rate was lower than 20%. Those variables that could not be imputed were excluded from the analysis (see Section 4).

### 2.1. Descriptive Analysis

We included a total of 952 individuals with CVD after quality controls. The clinical and demographic baseline characteristics from the sample are shown in Table 1. The mean age was 72.7, with a mean Age-A of 1.1 years. As this is a cohort of CVD individuals, there was a high prevalence of VRF (74% hypertension, 37% diabetes, 41% hyperlipidemia, 31% atrial fibrillation (AF), and 30% smokers). In the bivariate analysis (Table 2), we found that C-Age, sex, baseline modified Rankin Scale (mRS), smoking, alcoholism, drug consumption, weight, height, body mass index (BMI), ischemic heart disease, AF, leukocytes, neutrophils, triglycerides, and high-density lipoprotein (HDL) levels were significantly associated with B-Age, so we adjust the linear regression model for all of them. Beta coefficients assigned to each variable in this model are displayed in Appendix A. Note that the significant association between C-Age and Age-A is negatively correlated. This is why we include C-Age in the models. As has been previously reported, this association is mainly attributed to the “survival bias” effect, which, in stroke cohorts, is even more notorious. It basically refers to the fact that high Age-A is associated with higher mortality. Thus, older people (in C-Age) are individuals who survived when they were young, and are probably people with a lower Age-A, while people with high Age-A are overrepresented among young individuals because they are less likely to grow old.

### 2.2. Model Training

Our sample of patients with CVD was split into a training set and a test set, containing 25% and 75% of cases of the original cohort, respectively. Using the training set, we fitted six different models: a conventional linear regression model, four ML models (elastic net regression (EN), K-Nearest neighbors (KNN), random forest (RF), and support vector machine with radial kernel (SVM) models), and one DL approximation (multilayer perceptron (MLP) model). Hyperparameters were selected by applying a grid search via a 10-fold cross validation with five repeats, and we show the best configuration of parameters for each model in Appendix A. The final variables included in the EN regression model and their beta coefficients are shown in Appendix A. Regarding the RF model, we plotted the importance of each variable as the mean decrease in Gini (Figure 1). We observed that the most important variable in this model was C-Age, followed by leukocytes, neutrophils, triglycerides, and height. In addition, lymphocytes, BMI, and weight showed also a mean decrease in Gini above the average importance. On the other hand, lipids (low-density lipoprotein (LDL), HDL, and total cholesterol) showed an importance close to the average. Finally, other variables such as sex, VRF, or lifestyle factors showed less importance in the context of RF training.

In Figure 2 we plotted the average performance and variability of each model. We can see that in the training stage, there was a relatively high variability in all cases, but this was especially high for SVM and MLP models. We also can see that the two models showing a better performance in the training set (higher R^2^ and lower root mean squared error (RMSE) and mean absolute error (MAE) values) were the linear regression and EN models; although, all models, except for KNN, which showed the poorest performance, presented similar R^2^ values ranging from 0.32 to 0.35 in the training (Figure 2).

### 2.3. Model Evaluation

The performance of each model in the test set is shown in Table 3. We observed again that the KNN model showed the poorest performance (R^2^ = 0.27), but the rest of the models showed similar R^2^ values ranging from 0.34 to 0.38. The best-performing models were EN and MLP; although, the predictive capability was modest, such that MAE was 4.263 in the MLP (best-performing model), and the intraclass correlation coefficient (ICCC), a measure of concordance between observations, was 0.52 (Table 3), suggesting only a moderate agreement. These results are also visually displayed in Appendix A, where we see that the correlation between actual and predicted Age-A values is very similar within models, especially in the test set. We compared predictions and residuals obtained from EN and MLP models, and we observed a high correlation in both cases (r_predictions_ = 0.98, r_residuals_ = 0.99; *p*-value < 0.0001 in both cases), suggesting that both models arrived at similar predictions and lacked the same information (Figure 3).

### 2.4. Model Interpretation

As we only explained about 35.8% to 37.8% of Age-A variance in the best performing models (EN and MLP, respectively), we checked whether residuals (difference between real observations and predictions) from each model were associated with any of the variables that could not be imputed due to a high proportion of missing cases (see Section 4.5.2). These variables included diet habits, physical activity data, brain parenchymal fraction (BPF), some serum parameters (creatinine, estimated glomerular filtration rate (eGFR), glycosylated hemoglobin (HbA1c), and C-reactive protein (CPR)), and heart ejection fraction. However, we did not find any significant association between them with the residuals (Appendix A). We then analyzed the correlation between these variables with the predictions of both models, and many of them were significantly correlated, which suggests that our models already include the information that might be provided by these factors and there is some degree of overlap (Appendix A).

Finally, we evaluated the correlation between CpGs β-values used to calculate B-Age and the residuals of our MLP and EN models. We found that most of them were significantly correlated with residuals, confirming that residuals contain DNAm information used to estimate Age-A that, by definition, is not explained by the models, and thus not explained by VRF, lifestyle habits, and basic demographic data (Appendix A).

## 3. Discussion

This is the first study that provides information about whether clinical factors can predict Age-A based on DNAm data in patients with CVD. We could only explain between 35.8% and 37.8% of Age-A using the EN or MLP models, respectively. We explored whether other variables that we collected in our cohort regarding dietary or physical activity habits, or target organ damage (TOD) (chronic kidney disease, creatinine, or BPF) were associated with the residuals from these models, and we observed no significant correlations. This suggests that, at least in our sample, these factors do not provide additional information in the estimation of Age-A to the clinical factors already included in our models. However, this lack of association should be considered with caution, due to the small proportion of patients with information about these lifestyle factors.

Nonetheless, we confirmed previous findings that correlate biological aging with lifestyle factors, such as smoking, alcohol consumption, or BMI [12,29,30]. Moreover, we also confirm associations with other non-modifiable factors such as sex and height, which previously had been associated with increased epigenetic aging [29], age-related diseases, and mortality [31,32]. VRF are also known to be related to age, DNAm changes, and B-Age [33]. It has been described, for example, a significant correlation between accelerated epigenetic age and diabetes or incident AF [34,35]. Moreover, it is also known that there are age-related changes in the immune system featuring immunosenescence and chronic low-grade inflammation [36], which would explain why immune response factors included in our study (such as leukocytes) also contribute to the performance of the models.

Previous studies have demonstrated that not only are there lifestyle factors associated with B-Age, but that their modification can actually result in changes in DNAm and Age-A. For example, it has been reported that the effect of smoking on blood DNA methylation is partially reversible upon smoking cessation for longer than 3 months [37]. Additionally, Wu et al. found that, after smoking ceased, the epigenetic age acceleration in airway cells slowed to a level of that of non-smokers [38]. These findings reinforce the theory that at least part of aging can be easily modified.

However, these more accessible variables seem not to be enough to explain B-Age, given that our models explained less than 40% of the Age-A in the cohort. Although the interpretation of these results must be cautious, it seems that a relatively large proportion of the differences in Age-A is explained by other unknown factors, perhaps by socio-economic level, pollution, or other comorbid diseases. Another interpretation could be that an important part of Age-A is genetically determined and non-modifiable. There is increasing interest in what epigenetic clocks really represent [39]. While some researchers argue that epigenetic clocks might reflect the progression of regulated processes that are evolutionarily conserved across mammals, other researchers think that epigenetic clocks reflect just the results of stochastic damage in the DNA, which induces DNA methylation [40]. That is, whether these epigenetic scores have a causal role in aging, or they just represent a concomitant biomarker of specific biological processes. Moreover, many lifestyle, environmental, and cardiovascular factors are known to affect DNAm and B-Age, and indeed, we observed many of these associations in most of our univariate and multivariate analyses [40,41]. Hence, equivalent increased Age-A between patients might be a common consequence of their exposure to different factors. Altogether, it is challenging to construct a formula to estimate epigenetic Age-A based on phenotypic data; although, it is worth to still studying this possibility, especially in the light of newer epigenetic clocks that show a very good performance in predicting mortality risk [42]. This would help the use of Age-A estimations based on ML models in clinical practice, allowing the identification of patients at an increased risk of mortality that could undergo more strict clinical follow-ups to monitor VRF or medication compliance. However, the modest performance in the prediction of the presented models also has great relevance, indicating that there is still extra information captured in B-Age that cannot merely be explained with easily available clinical or epidemiological variables, which reinforces the utility of determining this epigenetic B-Age.

In our study, we tested six different mathematical models in the prediction of epigenetic Age-A. The use of ML and DL techniques only showed a small improvement compared to a regular approach. This is probably explained by the modest predictive capacity of our model, which, in a situation with noise a linear approximation, tends to be enough [43]. These approximations will be probably more useful in future studies combining phenotypic clinical data with other sources of data such as pollution exposures, genetic predisposition, or radiological techniques [44]. Adding more sources of data and using larger sample sizes based on multicentric cohorts may also improve ML predictions.

One valuable strength to be highlighted in the study is the accuracy in the acquisition and pruning of phenotyping data, given that the registers of the cohorts are based on clinical assessment and face-to-face visits by healthcare staff, instead of using administrative databases. This very fine phenotyping provides the study with a high reliability, which is essential for machine learning approaches [45].

The use of a CVD cohort is also an interesting point, given the characteristics of age and risk factors in this population. CVD is an age-related disease, so this population is older, with a higher prevalence of risk factors, and most likely to be under the effect of them for more years. Moreover, it has been reported that stroke patients, of the same C- Age, have higher biological ages than healthy individuals [46]. Therefore, this population may enhance our capability to detect the contribution of these factors to Age-A. However, it would be interesting to replicate this finding in a healthy cohort, since the epigenetic changes that occurred as a consequence of the CVD itself could have affected our results.

There are some limitations that must be considered. First, despite our sample size seeming to be appropriate for the models we created, it could be inadequate if we intend to create more extensive models, expand the number of variables, or include genetic information, environmental pollution, etc. This should be taken into consideration in the design of future studies. Moreover, for these future studies, it would be potentially interesting to train these models in larger multicentric cohorts to improve not only the performance, but also the translatability of our results. Second, DNA methylation was measured in peripheral blood cells. Although the Hannum algorithm is specifically designed for age estimation in blood samples, it is possible that biological age in other tissues could slightly differ, given that methylation of certain CpGs is tissue-specific [47]. Lastly, the study is not designed to establish causality between each variable and Age-A, so our results should not be interpreted at the individual level, nor in a cause–consequence way.

In summary, ML models could explain less than 40% of the variability in Age-A, which indicates that there is still relevant information captured in B-Age that cannot be explained by easily available clinical and epidemiological data. This reinforces the utility of determining the epigenetic B-Age. However, it would be worth trying to improve these predictive models, using more data, such as genetics, detailed diet habits, pollution exposures, lifestyle habits, and TOD, which would require larger samples and different cohorts, to better understand the underlying mechanisms of aging.

## 4. Materials and Methods

### 4.1. Setting

The study population came from the BASICMAR Register (consecutive patients assessed in Hospital del Mar from 2009 to 2018 with diagnosis of stroke (PI0517387) [34,48]. Inclusion criteria for this study were as follows: (1) diagnosis of acute CVD, including ischemic stroke, transient ischemic attack, or non-traumatic parenchymal intracranial hemorrhage; (2) availability of DNAm data; and (3) absence of concomitant illness such as neoplasm, demyelinating and autoimmune diseases, and vasculitis. All patients were assessed and classified by a neurologist and were included in the study by consecutive order of recruitment. A total of 952 patients met these criteria.

### 4.2. Ethics

Data collection for the study followed local research ethics guidelines. The identity of individual patients was completely anonymized. The study was approved by the local ethics committee (CEIC-Parc de Salut Mar, Barcelona, Spain). All participants or their approved proxy provided written informed consent for participation. The study was conducted according to the principles expressed in the Declaration of Helsinki and the applicable national legislation.

### 4.3. Clinical Variables

We included in our study all easily measurable variables available in our registry related to either age or lifestyle, such as information on VRF, diet, physical activity, and TOD.

#### 4.3.1. Vascular Risk Factors

VRF, defined as recommended by international consensus, were recorded in a direct interview with the patient, relatives, or caregivers, as well as from medical records, as previously reported [34]. Examinations were performed and standardized questionnaires administered during the hospitalization by a team of neurologists and reviewed by an additional neurologist. These data included C-Age, sex, weight, height, BMI, waist diameter, self-reported smoking habit, drug and alcohol consumption, AF, arterial hypertension, ischemic heart disease, diabetes mellitus, and hyperlipidemia. Glycemia, levels of total cholesterol, HDL and LDL cholesterol, creatinine, HbA1c, CRP, and leukocytes (including neutrophils, lymphocytes, and monocytes) were obtained via laboratory determinations at Hospital del Mar. Previous functional status was assessed through mRS.

#### 4.3.2. Lifestyle and Diet

Dietary and physical activity habits were available in a subgroup of consecutive individuals that were evaluated using three validated questionnaires: food frequency (FFQ) [49], Mediterranean Diet Score, and physical activity [50]. Energy consumption and total nutrient intake were calculated from the validated FFQ questionnaire using the MediSystem 2000 software (Sanocare Human Systems L.S., Madrid, Spain) [51].

#### 4.3.3. Target Organ Damage

Some markers of TOD were available in a subgroup of patients. Regarding the kidney, we calculated the estimated glomerular filtration rate (eGFR) using the CKD-EPI formula [52].

Heart ejection fraction was measured through echocardiogram.

Regarding brain TOD, magnetic resonance imaging studies (MRI) were available in a subset of patients. In those, we measured the brain parenchymal fraction (BPF), a well-established indicator of whole-brain atrophy. To calculate BPF, axial T1-weighted images were processed using a previously described neuroimaging pipeline [53]. Briefly, this workflow conducts a series of steps to label the native space from each patient according to an anatomical map. Hence, after correcting the bias field and performing the skull-stripping, several anatomical maps are non-linearly registered to patients’ native space using a robust method followed by a label fusion and intensity-based label refinement using expectation-maximization [53]. For this study, we only used data about the tissue-class segmentation (grey matter, white matter, and cerebrospinal fluid) due to the voxel anisotropy (clinical scans). We then calculated the BPF as the ratio between the number of voxels corresponding to parenchymal tissue (grey and white matter) and total number of voxels corresponding to the skull-stripped region, which also includes cerebrospinal fluid voxels:BPF = (Voxels _GM_ + Voxels _WM_)/(Voxels _GM_ + Voxels _WM_ + Voxels _CSF_)

### 4.4. Age Acceleration Estimation

#### 4.4.1. Array-Based DNA Methylation Quantification

DNA was extracted from whole-blood samples collected in 10 mL EDTA tubes at hospital arrival. Genome-wide DNA methylation was assessed using Infinium Human Methylation 450 BeadChip (Illumina, San Diego, CA, USA), interrogating 485,577 probes, and Infinium Methylation EPIC BeadChip (Illumina, San Diego, CA, USA), interrogating 865,918 probes. We followed the manufacturer’s protocol in all cases.

Data from both BeadChips were separately pre-processed using the same standardized pipelines [54,55]. Methylation values were corrected for background values and then normalized by SWAN using the minfi Bioconductor package [56]. Initial quality control of sample data was conducted using GenomeStudio version 2011.1 (Illumina, San Diego, CA, USA). The rest of the quality controls and sample analyses were performed as described in Soriano-Tarraga et al. [34]. The quality control of the raw methylation data was previously described [57,58]. Methylation status at each CpG site was reported by β-values, which range between 0 (completely unmethylated) and 1 (completely methylated).

#### 4.4.2. Biological Age

B-Age was calculated from DNA methylation levels of whole-blood samples. We chose the Hannum estimator for developing the whole study because it is designed for blood DNAm and it performed better in our blood samples [35]. This B-Age is calculated by summing the β-values of 71 specific CpGs multiplied by the effect size coefficient reported by Hannum et al. [14]. Six of the seventy-one CpGs used in this estimator are not included in the last Infinium Methylation EPIC BeadChip. However, in previous works, an excellent correlation was shown between the estimations made only with the 65 CpGs and those made with 71 CpGs [59], demonstrating that EPIC can be used for Hannum’s estimation.

We then calculated the individual Age-A by subtracting C-Age from B-Age [60]. Positive values correspond to individuals with higher B-Age than expected, and negative values to individuals with lower B-Age than expected. For our purpose of creating explanatory models, we chose Age-A as a dependent variable and included C-Age in the models.

### 4.5. Statistical Analyses

#### 4.5.1. Descriptive Statistics

Data were reported as mean (±standard deviation (SD)), median (interquartile range), or count (percentage). After assuming a normal distribution for Age-A, we studied the bivariate associations between Age-A and predictors of interest using Pearson’s correlation coefficient, *t*- or ANOVA tests according to the type and distribution of each variable.

#### 4.5.2. Missing Data, Imputation, and Data Pre-Processing

After selecting patients, we had a sample of 952 patients. However, many subjects presented missing data in some of the predictor variables. These missing data are reported in Appendix A. As machine learning techniques are highly dependent on sample size, we imputed those predictor variables that were missing in less than 20% of subjects. Therefore, we used a KNN-based imputation algorithm to impute: smoking status, alcoholism, drug consumption, height, weight, leukocytes, neutrophils, lymphocytes, monocytes, total cholesterol, HDL, LDL, and triglycerides [61]. Those variables that were missing in more than 20% of subjects were not imputed and, thus, were excluded for their use in model training: diet variables, physical activity, waist circumference, HbA1c, creatinine, CRP, eGFR, heart ejection fraction, and BPF.

Hence, after imputing missing data, we used the following variables to build a model for Age-A prediction: C-Age, sex, weight, height, BMI, white cell counts obtained from laboratory determinations (leucocytes, neutrophils, lymphocytes, and monocytes), lipids (total, and HDL and LDL cholesterol), VRF (hypertension, diabetes, hyperlipidemia, and atrial fibrillation), previous ischemic cardiopathy, smoking status, alcoholism, drugs consumption, and previous functional status.

We accurately checked outlier cases and the distribution in this set of variables. A senior Neurologist (IF-P) manually reviewed all cases showing at least one variable above or below 3 SDs from the sample mean and corrected these cases when necessary. Moreover, those continuous variables that presented a left-skewed distribution were log-transformed to achieve a normal distribution [43]. We only needed to log-transform white cell counts, triglycerides, and BMI. Finally, all variables were centered to 0 and the SD was scaled to 1, prior to using them in any model.

#### 4.5.3. Model Training

In this manuscript, we compared the Age-A predictions of 6 different models. A conventional linear regression model, 4 ML models (EN, KNN, RF, and SVM with a radial kernel), and 1 DL approximation (MLP model). For all models, we split the sample into 75% training and 25% test sets, and the same partition was used for all models.

Regarding the regular linear regression model, we adjusted it for the variables that were associated with Age-A in the bivariate analysis (see Section 4.5.1) with a *p* value less than 0.05.

For ML and DL models, hyperparameters were selected using a grid search based on a 10-fold cross validation with 5 repeats in the training set. Objective function was the RMSE in all cases. The model configuration showing the lower RMSE was selected for evaluation in the test set. For ML models (EN, KNN, RF, and SVM with a radial kernel) we used the Caret library in R, while for the MLP model, we used the tensor flow library in Python [62,63]. In Appendix A, we show which parameters were tuned in each model.

Moreover, for the EN model, we applied a polynomial expansion to allow the modeling of non-linear associations. This is undertaken by considering the squared and cubic transformations from all continuous variables during the model training. For the RF model, we also checked the importance of each variable as the mean decrease in Gini (quality of the splits) when that variable was excluded from the model. Those variables having a higher Gini index are providing more information to the model.

#### 4.5.4. Model Evaluation

We used several metrics to compare the performance of the models mentioned above in the test set. Specifically, models were evaluated by assessing the R^2^, RMSE, MAE, and the ICCC, which is a metric representing the concordance between observations [64].

#### 4.5.5. Model Interpretation

After comparing models, we selected those with the best performance (EN and MLP). We then calculated the Age-A predictions residuals, as the difference between Age-A predictions and actual values in the whole sample. To achieve further insight into what our models were representing, those variables that were not included in the models’ training due to an elevated proportion of missing cases (see Section 4.5.2), were correlated with both models’ predictions and residuals. Correlation with predictions might provide insight into which pathological processes are associated with these estimations, while correlations with residuals might give direction on which variables might contribute to explaining the variance that is not currently explained by the models.

## 5. Conclusions

We constructed different ML models to evaluate the influence of easily measurable clinical and lifestyle factors on Age-A in a CVD cohort. The best model (MLP) was able to explain 38% of the variability in Age-A, supporting the influence of these factors in the process, but also suggesting that they are not enough to explain most of its variability.

## Figures and Tables

**Figure 1 ijms-24-02759-f001:**
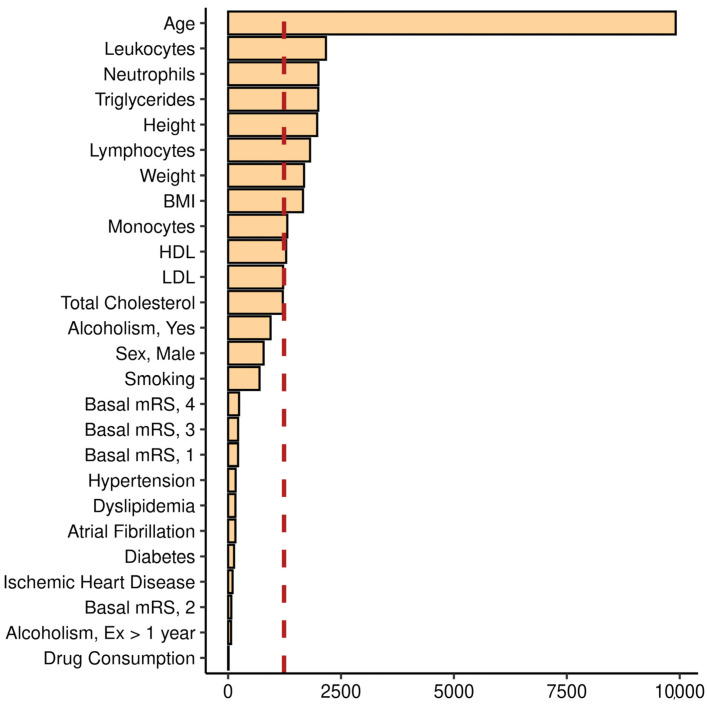
Importance plot obtained from Random Forest model training. The *y* axis shows one tick per variable, while the *x* axis shows the mean decrease in Gini index (purity of splits). The red dashed line shows the average importance of the variables selected for the training. Keywords: BMI, body mass index; HDL, high-density lipoprotein; LDL, low-density lipoprotein; mRS, modified Rankin scale; Ex, ex-alcohol consumption.

**Figure 2 ijms-24-02759-f002:**
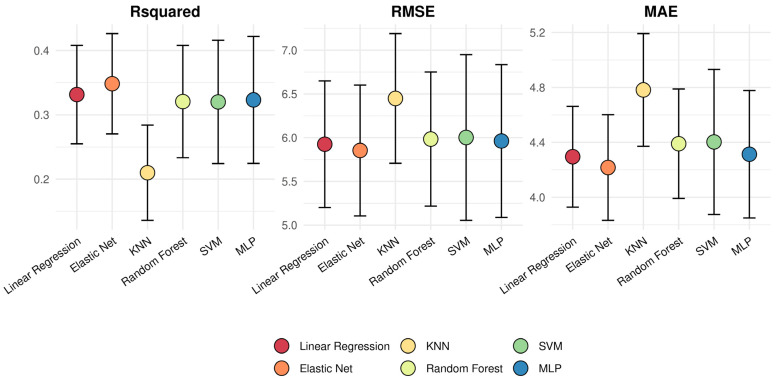
Variability in the performance of models’ training. In this plot, we show the variability in each metric of interest (R^2^, RMSE, and MAE) of the best model configuration obtained via 10-fold cross validation (5 repeats). Dots represent the average performance and error bars correspond to the standard deviation from each metric distribution. Keywords: MAE, mean absolute error; RMSE, root mean squared error; KNN, K nearest neighbors; SVM, support vector machine; MLP, multilayer perceptron.

**Figure 3 ijms-24-02759-f003:**
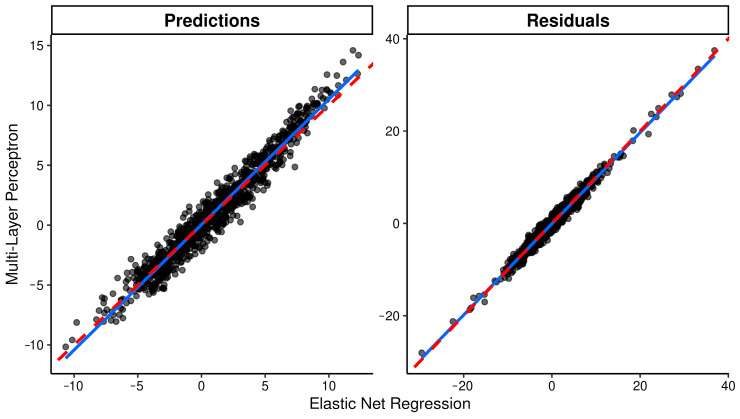
Comparison between elastic net regression and multilayer perceptron models. In this figure, we show the correlation between elastic net regression and multilayer perceptron models’ predictions and residuals. Each dot represents the value in one individual of the test set, the red-dashed lines represents the ideal correlation and the blue lines represent the real correlation in the data. High correlations suggest a certain degree of similitude between models.

**Table 1 ijms-24-02759-t001:** Main characteristics from the study sample (N = 952).

Demographic Variables	
Age, years	72.68 (±12.4)
Biological Age, years	73.76 (±10.4)
Age Acceleration, years	1.081 (±7.3)
Sex, Female	404 (42.4%)
**Morphometric Measurements**	
Weight, kg	73.3 (±13.4)
Height, cm	163.7 (±9.2)
BMI, kg/m^2^	26.7 (±4.6)
**Vascular Risk Factors**	
Hypertension	701 (73.6%)
Diabetes	352 (37.0%)
Hyperlipidemia	448 (47.1%)
Ischemic heart disease	128 (13.4%)
Atrial fibrillation	293 (30.8%)
**Laboratory Determinations**	
Leukocytes, u/mcL	8917 (±2937)
Neutrophils, u/mcL	6285 (±2732)
Lymphocytes, %	21.8 (±9.3)
Monocytes, %	6 (±2.2)
Total cholesterol, mg/dL	174.2 (±41)
Triglycerides, mg/dL	124.5 (±63.2)
HDL, mg/dL	47.1 (±12.5)
LDL, mg/dL	103.1 (±34.2)
**Lifestyle**	
Active smoker	284 (29.8%)
Alcoholism	
No	673 (70.7%)
Previous alcoholism > 1 year	42 (4.41%)
Yes	237 (24.9%)
Drug consumption	22 (2.31%)
**Previous Functional Status**	
Baseline mRS	
0	611 (64.2%)
1	113 (11.9%)
2	100 (10.5%)
3	93 (9.8%)
4	33 (3.5%)
5	2 (0.2%)

Values represent mean (±SD) or number (percentage) depending on the type of each variable (continuous or categorical). Keywords: BMI, body mass index; HDL, high-density lipoprotein; LDL, low-density lipoprotein; mRS, modified Rankin scale.

**Table 2 ijms-24-02759-t002:** Bivariate associations between predictors of interest and age acceleration.

Variable	Correlation^®^	Average Age-A (±SD)	*p*-Value
Demographic Variables			
Age, years	−0.55		<0.001 *
Biological Age, years	0.05		0.1273
Sex:			<0.001 *
Female		−1.22 (±6.6)	
Male		2.78 (±7.27)	
Morphometric Measurements			
Weight (kg)	0.23		<0.001 *
Height (cm)	0.3		<0.001 *
BMI (kg/m^2^)	0.07		0.02 *
Vascular Risk Factors			
Hypertension			0.78
No		1.19 (±7.23)	
Yes		1.04 (±7.28)	
Diabetes			0.952
No		1.07 (±7.39)	
Yes		1.1 (±7.06)	
Hyperlipidemia			0.229
No		0.81 (±7.60)	
Yes		1.38 (±6.86)	
Ischemic heart disease			0.01 *
No		1.3 (±7.34)	
Yes		−0.36 (±6.59)	
Atrial fibrillation			<0.001 *
No		1.98 (±7.26)	
Yes		−0.93 (±6.87)	
Laboratory Determinations			
Leukocytes, u/mcL	0.14		<0.001 *
Neutrophils, u/mcL	0.13		<0.001 *
Lymphocytes, %	−0.02		0.517
Monocytes, %	−0.004		0.898
Total cholesterol, mg/dL	−0.008		0.803
Triglycerides, mg/dL	0.12		<0.001 *
HDL, mg/dL	−0.15		<0.001 *
LDL, mg/dL	0.05		0.0868
Lifestyle			
Smoking			<0.001 *
No		−0.29 (±6.93)	
Yes		4.31 (±7.00)	
Alcoholism			<0.001 *
No		−0.19 (±7.22)	
Previous alcoholism > 1 year		1.57 (±5.35)	
Yes		4.6 (±6.49)	
Drug consumption			<0.001 *
No		0.97 (±7.27)	
Yes		5.88 (±4.92)	
Previous Functional Status			
Baseline mRS	−0.22		<0.001 *

Average Age-A column represents the average value of age acceleration for each category in categorical variables. The correlation^®^ column indicates the correlation coefficient between Age-A and the continuous variable. The last column shows the *p*-values testing the association between each predictor of interest and Age-A (Pearson’s correlation coefficient, *t*- or ANOVA test depending on the type of each variable), and the * marks statistically significant values. Keywords: BMI, body mass index; HDL, high-density lipoprotein; LDL, low-density lipoprotein; mRS, modified Rankin scale.

**Table 3 ijms-24-02759-t003:** Performance of each model in the test dataset.

	r	R²	RMSE	MAE	ICCC
Linear Regression	0.583	0.34	5.969	4.287	0.51
Elastic net	0.598	0.358	5.917	4.248	0.49
K nearest-neighbors	0.516	0.266	6.401	4.747	0.37
Random Forest	0.579	0.335	5.992	4.371	0.49
Support Vector Machine	0.584	0.341	6.083	4.467	0.44
Multi-Layer Perceptron	0.615	0.378	5.852	4.263	0.52

Values represent correlation coefficients (r), determination coefficient (R^2^), root mean squared error (RMSE), mean absolute error (MAE), and intraclass correlation coefficient (ICCC). Higher values indicate a better performance for r, R^2^, and ICCC metrics, while lower values indicate better performance for RMSE and MAE.

## Data Availability

Data will be shared with qualified researchers upon request.

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
