# Peer review of "Machine Learning Approximations to Predict Epigenetic Age Acceleration in Stroke Patients"

_ijms, 2023, doi:10.3390/ijms24032759_

Round 1
Reviewer 1 Report
The authors present a study in which they attempt to build ML models to predict age acceleration based on DNAm data (estimation of biological age) and other clinical factors. While they found trends in the data that concur with others in the literature, they were not able to develop a model that significantly predicts age acceleration - in that the features that were assessed did not explain the variability in the data.
I have several comments that can be addressed that can help in the understandability of the paper:
Please explain in more detail the relationship between the DNAm data and biological age.
Please include more narrative throughout the results of the paper. Since the methods is after the Discussion, this would help the reader understand the flow of logic.
Please explain in more detail the contribution of the work given that the r values were so small (albeit significant).
What could be done to make ML models that predict Age-A better?
Reviewer 2 Report
The manuscript "Machine Learning Approximations to Predict Epigenetic Age Acceleration in Stroke Patients" - constitutes a useful scientific approach to show the influence of environmental factors, lifestyle, and vascular risk in the acceleration of aging, using different machine learning strategies. The study involved a cohort of 952 stroke patients. The acceleration of aging was based on Hannum's epigenetic clock, and 6 different models were trained to predict the acceleration of aging. The improvement of the article could target at least the following aspects:
1. Age Acceleration (Age-A) is a useful tool capable of predicting a wide range of health outcomes – as the abstract begins and then the idea is repeated in the introduction, it is a very general statement that denotes a somewhat superficial approach without strong scientific support. Specifically, which, how, and in what way can we consider accelerated aging to be predictive of which aspect of health or mortality, and what is the scientific evidence (with references);
2. regardless of the molecular and genetic factors that may contribute to accelerated aging, the burden of environmental, lifestyle, and vascular risk (VRF) factors is known to affect this process. The statements are somewhat superficial and would require a more concrete expression and a much more inclined approach, considering the fairly consistent scientific literature in this field.
3. The aim of this study was to estimate the contribution of easily measurable clinical factors to accelerated aging in patients with cerebrovascular disease (CVD) to investigate possible mechanisms involved in accelerated aging. The question which I would first ask is if we should not start from a basic model without cerebrovascular disease - normal aging.
4. The parameters considered are very different. Is there a certain rationale for their selection?
5. It is indeed challenging to construct a formula for estimating epigenetic aging based on phenotypic data, and it is certainly worth studying this possibility further, especially in light of newer epigenetic clocks that show very good performance in predicting mortality risk. In this case, what would be the target to be reached from the point of view of the developed artificial intelligence models;
6. Sample size tends to be inadequate for building models, which usually require robust and large amounts of data. It would be necessary to train these models on a larger cohort to improve their performance. What might be the way to determine the appropriate sample size?
7. The conclusion of the study does not denote the importance and scientific relevance of the study to the extent that the expectations in the title are much higher. The explanation of the obtained results, the existing limitations, the conditions generated by the respective models, and the future approaches in the field should be emphasized.
Round 2
Reviewer 2 Report
The authors responded very consistently to all my previous comments. As the responses fulfill my improvement suggestions, my recommendation is for acceptance in the present form.